# MVoice: Multilingual Unified Voice Generation With Discrete Representation at Scale

## Abstract

Various applications of voice synthesis have been developed independently despite the fact that they generate "voice" as output in common. In addition, the majority of voice synthesis models currently rely on annotated data, but it is crucial to scale them to self-supervised datasets in order to effectively capture the wide range of acoustic variations presented in human voice, including speaker identity, emotion, and prosody. In this work, we propose MVoice, a multimodal spoken large language model for synthesizing and manipulating voice signals at scale. MVoice employs self-supervised voice tokens with the "coarse-to-fine" designs to first determine semantic meaning and then introduce condition signals for acoustic generation. It offers notable benefits with unified generation and transformation capabilities: 1) model and data scalability: without the requirement of scattered model-specific methodologies or annotations acoustic data, training could be scaled up in terms of data usage and model capability; and 2) controllability and conditioning flexibility: we investigate different conditioning mechanisms and effectively handle voice synthesis applications, including text-to-speech, voice conversion, singing voice synthesis, singing voice conversion, and speech-to-speech translation by re-synthesizing the discrete representations with prompt guidance. Experimental results demonstrate that MVoice exhibits superior audio quality and style similarity compared with competitive baseline models in monolingual/cross-lingual voice generation. [1]

## 1 Introduction

Table 1: Supported audio generation tasks of MVoice and selected prior works. The versatile MVoice model supports more tasks than all others: it supports 5 voice generation tasks in 6 languages. ML: Multilingual; TTS: text-to-speech; VC: voice conversion; SVS: singing voice synthesis; SVC: singing voice conversion; S2ST: speech-to-speech translation;

| Model | Feature | ML | TTS/VC | SVS | SVC | S2ST |
|---|---|---|---|---|---|---|
| VoiceBox (Le et al., 2023) | Continuous | ✓ | ✓ | ✗ | ✗ | ✗ |
| NaturalSpeech2 (Shen et al., 2023) | Continuous | ✗ | ✓ | ✓ | ✗ | ✗ |
| Maga-TTS (Jiang et al., 2023) | Continuous | ✗ | ✓ | ✗ | ✗ | ✗ |
| VALL-E X (Zhang et al., 2023) | Discrete | ✓ | ✓ | ✗ | ✗ | ✓ |
| SPEAR-TTS (Kharitonov et al., 2023) | Discrete | ✗ | ✓ | ✗ | ✗ | ✗ |
| SpeechX (Wang et al., 2023b) | Discrete | ✓ | ✓ | ✗ | ✗ | ✗ |
| MVoice (Ours) | Discrete | ✓ | ✓ | ✓ | ✓ | ✓ |

Voice synthesis (Wang et al., 2017; Ren et al., 2019; Qian et al., 2020) aims to generate human-like voices, which attracts broad interest in the machine learning community. These voice generation models have been extended to more complex scenarios, including multiple speakers, emotions, and styles for expressive and diverse voice generation. A growing number of applications, such as voice assistant services and long-form reading, have been actively developed to real-world speech platforms.

---

[1]Audio samples are available at `https://MVoice.github.io`

Despite the recent success of deep generative models (Casanova et al., 2022; Huang et al., 2022b; Min et al., 2021; Nachmani & Wolf, 2019), the rising demand for expressive voice generation poses challenges for models in 1) zero-shot generalization: when the distributions of testing samples differ from training data with unseen acoustic diversities (e.g., speaker identity, emotion and prosody), the quality of synthesized voice often deteriorates due to distribution gaps, and 2) unified generation: most of current voice models have been developed independently, where the methodologies developed for each application remain scattered in research fields.

In this work, we introduce MVoice, a multimodal large language model for synthesizing and manipulating multilingual voice signals at scale. MVoice employs self-supervised tokens with the following "coarse-to-fine" generation designs: 1) Semantic stage determines the desired meaning given text or speech; 2) Acoustic stage is scaled to a large amount of self-supervised audio-only data, where conditioning mechanisms are investigated with various control signals. The single decoder-only model is trained on a mixture of tasks that involve arbitrarily interleaved voice.

MVoice demonstrates notable advantages as a general-purpose unified model in voice synthesis: 1) Model and data scalability: without the requirement of annotation data or scattered model-specific methodologies, training could be scaled up in terms of data usage and model capability; 2) Controllability with flexible conditioning options: various conditioning mechanisms are investigated by re-synthesizing the semantic or acoustic representations with prompt-guided in-context learning.

MVoice is trained on ~200K hours of multilingual data in 6 languages, and we introduce 5 voice generation applications: text-to-speech (TTS), voice conversion (VC), singing voice synthesis (SVS), singing voice conversion (SVC), and speech-to-speech translation (S2ST). These applications can be effectively addressed by leveraging the unified framework that employs discrete representations. Experimental results demonstrate that MVoice achieves state-of-the-art results in monolingual/cross-lingual zero-shot voice generation. Both subjective and objective evaluation metrics show that MVoice exhibits superior audio quality and style similarity compared with baseline models. Our contributions can be summarized as follows:

- We propose a multimodal large language model called MVoice for unified voice synthesis, where the "coarse-to-fine" design aims to effectively model the human voice by considering semantic meanings and acoustic conditions.
- We investigate data and model scalability with 6 languages of multilingual data.
- We unify generation and transformation capabilities, allowing a mixture of 5 voice generation applications including text-to-speech (TTS), voice conversion (VC), singing voice synthesis (SVS), singing voice conversion (SVC), and speech-to-speech translation (S2ST).
- Experimental results demonstrate that MVoice achieves state-of-the-art results in monolingual/cross-lingual zero-shot voice generation. MVoice excels in scalability, controllability, and conditioning flexibility.

## 2 RELATED WORKS

Text-guided voice synthesis (text-to-speech and singing voice synthesis) typically converts input text into mel-spectrogram (e.g., Tacotron Wang et al. (2017), FastSpeech Ren et al. (2019)), which is then transformed to waveform using a separately trained vocoder (Kong et al., 2020; Huang et al., 2021), or directly generate waveform from text (e.g., EATS Donahue et al. (2020) and VITS Kim et al. (2021)). In zero-shot scenarios, when the distributions of style prompts deviate from the training data, the quality of the synthesized voice often suffers degradation due to distribution mismatches: GenerSpeech (Huang et al., 2022b) leverages multi-level style adaptors for the global and local stylization of the custom utterance. YourTTS (Casanova et al., 2022) is built upon VITS with several novel modifications for zero-shot multi-speaker and multilingual training. In this work, we enhance zero-shot robustness by scaling up training data with an extensive collection of speakers encompassing diverse accents, demographics, and recording conditions. This approach aims to capture the acoustic diversity presented in human speech, including variations of speaker identity, emotion, and prosody.

Beyond text-guided voice synthesis, speech-guided voice synthesis has made massive progress to date. Voice conversion (VC) (Liu et al., 2021; Qian et al., 2020; Yi et al., 2020) and singing voice conversion (SVC) (Deng et al., 2020; Nachmani & Wolf, 2019; Huang et al., 2023) aim to convert

only the speaker identity attribute while keeping semantic meaning the same. To replace timbre features extracted from referenced audio and keep content and pitch features unchanged, the key is decomposing the voice into timbre, pitch, and content representations. In contrast, speech-to-speech translation (S2ST) aims at converting speech from one language into speech in another, significantly breaking down communication barriers between people not sharing a common language. Direct systems (Lee et al., 2021a;b) leverage recent progress on self-supervised discrete units learned from unlabeled speech for building textless S2ST, and Popuri et al. (2022) show that self-supervised encoder and decoder pre-training with weakly-supervised data improves model performance.

Most aforementioned generative models are task-specific and trained on different datasets, while building a simple and unified voice synthesis framework has also attracted increasing attention in the community: Liu et al. (2023) make use of large amounts of unlabeled data for model training and boost the performance of zero-shot text-to-speech and voice conversion simultaneously. NANSY families (Choi et al., 2021a; 2022) are trained in a self-supervised manner that does not require any annotations paired with audio. Built-in continuous vector space, VoiceBox (Le et al., 2023) leverage non-autoregressive flow-matching model for speech at scale, and NaturalSpeech2 (Shen et al., 2023) uses a diffusion model to generate neural codec-based latent vectors conditioned on text input.

Another line of works models voice with an autoregressive transformer in a compact and discrete space. VALL-E (Wang et al., 2023a), SPEAR-TTS (Kharitonov et al., 2023) are proposed to clone a human's voice with discrete prompt tokens from a short recording (3-seconds). Speech-X (Wang et al., 2023b) combines neural codec language modeling with multitask learning using task-dependent prompting, which is capable of zero-shot TTS and various speech transformation tasks, dealing with both clean and noisy signals. MVoice has several design advantages compared to these baselines: 1) model and data scalability: without the requirement of annotations acoustic data or scattered model-specific methodologies, training could be scaled up in terms of data usage and model capability; 2) Controllability with flexible conditioning options: various conditioning mechanisms are investigated by re-synthesizing the semantic or acoustic representations with prompt guidance. MVoice exhibits in-context learning abilities by presenting how to create the context to perform tasks MVoice is not explicitly trained on.

## 3 MVOICE

### 3.1 OVERVIEW

MVoice is considered a unified voice synthesis model with a "coarse-to-fine" design that progressively enhances the modeling of voice signals by injecting desired conditioning information, which is organized in two main stages as illustrated in Figure 1: 1) semantic stage $S_1$, speech or text inputs are transformed into a sequence of semantic tokens $s$, 2) acoustic stage $S_2$, acoustic tokens $a$ with a variety of conditions (speaker, emotion, prosody, and style) are generated autoregressively from the "pseudo" text (i.e., semantic tokens $s$). In the end, a unit-based vocoder synthesizes high-fidelity waveforms from compressed acoustic representations.

### 3.2 VOICE REPRESENTATION

**Semantic tokens.** It is crucial to extract rich linguistic information from the speech signal. To this end, we resort to XLSR-53: a wav2vec 2.0 model pre-trained on 56k hours of speech in 53 languages (Conneau et al., 2020). In the following, a k-means algorithm is applied to the learned representations of the unlabelled speech to generate $K_1$ cluster centroids at every 20-ms frame. In the end, a speech utterance $y$ is represented as semantic tokens with $[s_1, s_2, \ldots, s_T], s_i \in \{0, 1, \ldots, K_1 - 1\}, \forall 1 \le i \le T$, where $T$ is the number of frames.

**Acoustic tokens.** The audio encoder $E$ of codec models (Zeghidour et al., 2021; Défossez et al., 2022) consists of several convolutional blocks with a total downsampling rate of 320 and generates continuous representations at every 20-ms frame in 16kHz. The residual vector-quantizer $Q$ produces discrete representations $a_q$ with a codebook size of $K_2$, using a vector quantization layer (Vasuki & Vanathi, 2006). In the end, a speech utterance $y$ is represented as acoustic tokens with $[a_1, a_2, \ldots, a_T], a_i \in \{0, 1, \ldots, K_2 - 1\}, \forall 1 \le i \le T$, where $T$ is the number of frames.

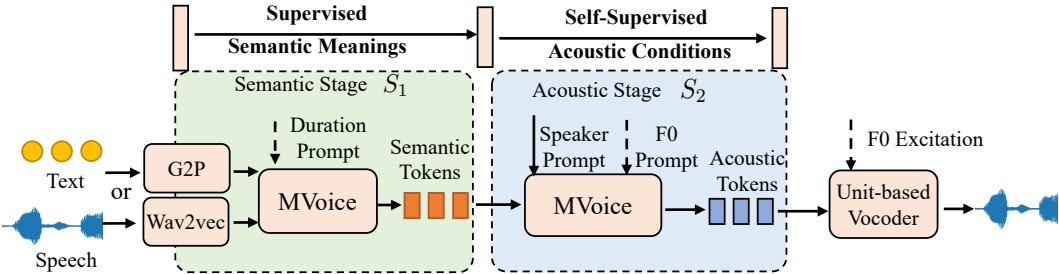

Figure 1: A high-level overview of MVoice. Note that $S_1$ and $S_2$ are learned jointly in a decoder-only language model. The F0 auxiliary input denoted with dotted lines is included only for singing voice.

### 3.3 MODEL: "COARSE-TO-FINE" DESIGN

#### 3.3.1 SEMANTIC STAGE $S_1$: DETERMINING SEMANTIC MEANING

Semantic stage $S_1$ maps tokenized text or speech into semantic tokens $s$. For semantic token modeling, we use parallel text-semantic or speech-semantic data to learn this mapping with an autoregressive decoder-only transformer architecture $\theta_{AR}$. It is conditioned on the condition signals $\mathbf{c_s}$ (multilingual phonemes), formulated as $p\left(\mathbf{s} \mid \mathbf{c_s}; \theta_{AR}\right) = \prod_{t=0}^{T} p\left(\mathbf{s}_t \mid \mathbf{s}_{<t}, \mathbf{c_s}; \theta_{AR}\right)$

#### 3.3.2 ACOUSTIC STAGE $S_2$: INTRODUCING ACOUSTIC CONDITIONS

Acoustic stage $S_2$ maps semantics into acoustic tokens $a$, where we flatten all the codebooks from the codec model. $S_2$ is scaled up to a large amount of self-supervised audio-only data containing many speakers with various accents, diverse demographics, and heterogeneous recording conditions to improve the robustness in zero-shot scenarios. It is designed to include a variety of acoustic conditions (e.g., speaker, emotion, prosody, and style) on top of semantic meanings, and thus we investigate different conditioning mechanisms for controllability and flexibility. It is conditioned on the condition signals $\mathbf{c_a}$ (semantic tokens, acoustic prompts, or explicit F0), formulated as $p\left(\mathbf{a} \mid \mathbf{c_a}; \theta_{AR}\right) = \prod_{t=0}^{T} p\left(\mathbf{a}_t \mid \mathbf{a}_{<t}, \mathbf{c_a}; \theta_{AR}\right)$

### 3.4 RECONSTRUCTING HIGH-FIDELITY WAVEFORMS

We train a unit-based neural vocoder from scratch for the acoustic unit to waveform generation. Inspired by BigVGAN (Lee et al., 2022), the synthesizer includes the generator and multi-resolution discriminator (MRD). The generator is built from a set of look-up tables (LUT) that embed the discrete representation and a series of blocks composed of transposed convolution and a residual block with dilated layers. The transposed convolutions upsample the encoded representation to match the input sample rate, while the dilated layers increase the receptive field. More details have been included in Appendix C.2.

### 3.5 ARCHITECTURE AND TRAINING

#### 3.5.1 ARCHITECTURE

With large-scale training data and powerful models, large language models have recently exhibited high-quality samples in natural language processing. To make audio modeling more tractable, recent studies propose to represent audio signals as multiple streams of discrete tokens representing the same signal and flatten these codes (Agostinelli et al., 2023; Kreuk et al., 2022). It comes at the high computational cost of modeling extremely long sequences, because of the quadratic cost of self-attention and large feedforward networks per-position.

To tackle the aforementioned issue, inspired by Yu et al. (2023), MVoice (denoted as $\theta_{AR}$) predicts long sequences with multiscale transformers. Specifically, 1) the token embedding matrix $E_G$ maps integer-valued tokens $x_{0..T}$ to $m$ dimensional embeddings, and concatenate with continuous speech representation in time axis (if any), following which 2) we chunk it into patches of size $P$ of length $K = \frac{T}{P}$, 3) a large global transformer $\theta_{AR}^{\text{global}}$ module outputs patch representations

$\mathbf{G_o^{1:K}} = \theta_{AR}^{\text{global}}(\mathbf{G_i^{0:K-1}})$, and 4) a small local transformer module operates on a single patch containing $P$ elements, each of which is the sum of an output from the global model and an embedding of the previous tokens, and autoregressively predict the next patch $\mathbf{L_o^{1:K}} = \theta_{AR}^{\text{local}}(\mathbf{L_i^{0:K-1}} + \mathbf{G_o^{1:K}})$. The model is optimized to maximize the probability of the next token, and we share the parameters of the output projection layer with the parameters of the embedding. This enables sub-quadratic self-attention and much larger feedforward layers for the same compute, unlocking the ability to train much larger and better-performing models and scale to very long audio sequences.

### 3.5.2 MULTILINGUAL AND MULTITASK TRAINING

**Scalability.** 1) Data: the acoustic modeling $S_2$ stage does not require any annotations, and thus training data could be scaled up to a large number of speakers, with various accents, diverse demographics, and heterogeneous recording conditions. It enables capturing acoustic diversity (speaker identity, emotion, prosody) in human voice, especially for zero-shot scenarios. 2) Model: Like GPT-3, large language models are all variants of the transformer architecture (Vaswani et al., 2017), where the improvements have primarily come from scaling the models' size in depth and width. Without the requirement of scattered model-specific methodologies, training could be scaled up regarding model capability.

**Languages and tasks module.** We signal to the model which language or task it should perform on a given input by prefixing the input with a tag specifying the task and language. For example, to query the model to perform text-to-semantic translation on an utterance in English, the tokenized input would be preceded by the two tags [T2S] [En]. To perform speech-to-semantic translation from English to French, the tokenized input would be preceded by [S2S] [En-Fr]. The component tasks that we consider in this report are 1) Text-to-semantic: mapping the phone sequence to semantic tokens; 2) Text-to-semantic with duration: mapping the phone sequence to semantic tokens given explicit duration guidance; 3) Speech-to-semantic: translating the speech sequence to semantic tokens; 4) Semantic-to-acoustic: translating the semantic sequence to acoustic tokens given acoustic prompt $\mathbf{a_p}$, and 5) Semantic-to-acoustic with $\mathbf{F_0}$: translating the semantic sequence to acoustic tokens given acoustic prompt $\mathbf{a_p}$ and explicit $\mathbf{F_0}$.

### 3.6 APPLICATION

MVoice exhibits competitive advantages as a unified voice synthesis framework with a "coarse-to-fine" design. As illustrated in Table 2, we demonstrate that MVoice exhibits in-context learning abilities by presenting how to perform tasks MVoice is not explicitly trained on.

**Zero-shot TTS / VC.** Given a target text $\mathbf{y}$, zero-shot TTS aims to generate high-quality speech samples with acoustic prompt $\mathbf{a_p}$ derived from a reference utterance, which has different acoustic conditions from training data. To control the characteristics of the speaker's voice, a prompt for timbre guidance is required. During training, we randomly select two non-overlapping windows of speech from each example, and consider one of the windows as the prompt and the other as the target output. Instead, we extract the semantic tokens from speech using the K-means model for voice conversion.

**Zero-shot SVS / SVC.** Given the frame-level phone $\mathbf{d}_f$, SVS aims to generate a singing voice with accurate pitch control, where the fundamental frequency $\mathbf{F_0}$ prompt is further requested in semantic-to-acoustic modeling. In practice, $\mathbf{F_0}$ could be predicted by a separately-trained neural network given MIDI score, and thus we directly take the $\mathbf{F_0}$ value as condition signals in acoustic model $S_2$ for simplification following (Liu et al., 2022), where we extract $\mathbf{F_0} = (f_1, \ldots, f_L)$ using the YAAPT algorithm (Kasi, 2002) from target singing voice with 320 hop size.

**Textless S2ST.** Unit-based textless S2ST system consists of a speech-to-unit translation (S2UT) model followed by a unit-based vocoder that converts discrete units to speech, we use the parallel data to learn this mapping from wav2vec continuous vector to the semantic token. Wav2vec 2.0 is a self-supervised framework to learn speech representations from unlabeled audio data, which is trained via contrastive loss with masked spans on the input to the context encoder. Shown in (Popuri et al., 2022), the self-supervised encoder pre-training with weakly-supervised data significantly improves model performance.

Table 2: Zero-shot applications by re-synthesizing the semantic or acoustic representations with varying conditions.

| Applications | $S_1$ | $S_2$ |
|---|---|---|
| TTS | Text-to-semantic | Semantic-to-acoustic |
| SVS | Text-to-semantic with duration | Semantic-to-acoustic with $\mathbf{F_0}$ |
| VC | K-means | Semantic-to-acoustic |
| SVC | K-means | Semantic-to-acoustic with $\mathbf{F_0}$ |
| S2ST | Speech-to-semantic | Semantic-to-acoustic |

## 4 DATA AND METRICS

### 4.1 DATASET

Table 3 lists the used datasets with six languages: **English (En), French (Fr), German (De), Spanish (Es), Japanese (Ja) and Chinese (Zh)**, and overall we have ∼**200k** hours 16 kHz audio as training data. For text sequence, we tokenize it into the phoneme sequence with an open-source grapheme-to-phoneme conversion tool (Sun et al., 2019). During the evaluation, we randomly choose sentences to construct the zero-shot testing set for each application task, in which the voice used for prompting is never seen by the model at training, and it has to reproduce the characteristics from a single prompt example. We have attached detailed information on the data configuration in Appendix A.

Table 3: Dataset usage in training and inference stages.

| Tasks | Language | Dataset | Testing set |
|---|---|---|---|
| TTS/VC | Ja, De, Fr, En, Es, Zh | Librilight, Gigaspeech, WenetSpeech, CSS, AISHELL | LibriTTS/VCTK |
| SVS | En, Zh | OpenSinger, M4Singer, CSD, Kiritan | Opencpop |
| S2ST | Fr, En, Es, De | SpeechMatrix | SpeechMatrix test |

### 4.2 TRAINING AND EVALUATION

**Intelligibility and accuracy.** We employ word error rate (WER) to evaluate the intelligibility of the generated speech by transcribing it using a wav2vec ASR system. We transcribe the translated speech for accuracy and then calculate the BLEU score (Papineni et al., 2002) between the generated and the reference text. For English-only setups, we use the large model pretrained and fine-tuned on Libri-Light and Librispeech on 16kHz sampled speech audio. For multilingual settings, we use ASR models publicly released on Hugging Face following (Duquenne et al., 2022).

**Style quality and similarity.** Speaker similarity score (SIM) assesses the coherence of the generated speech in relation to the speaker's characteristics, which is calculated as the cosine similarity between the speaker embeddings of the generated speech and the desired speech signals. F0 Frame Error (FFE) measures the timbre and prosody similarity of synthesized and reference audio, respectively.

**Subjective evaluation.** We also conduct a crowd-sourced human evaluation via Amazon Mechanical Turk, which is reported with 95% confidence intervals (CI), and analyze two aspects: style similarity (speaker, emotion, and prosody) and audio quality (clarity, high-frequency), respectively scoring SMOS and MOS. More information has been attached in Appendix D.

### 4.3 BASELINE

We compare the generated audio samples with other systems, including 1) GT, the ground-truth audio; 2) YourTTS (Casanova et al., 2022), GenerSpeech (Huang et al., 2022b), VALL-E (Wang et al., 2023a) for English zero-shot TTS; 3) YourTTS (Casanova et al., 2022) for zero-shot multilingual TTS; 4) NANSY (Choi et al., 2022) and PPG-VC (Liu et al., 2021) for VC; 5) Diffsinger (Liu et al., 2022) and FFT-Singer for SVS; 6) SpeechMatrix (Duquenne et al., 2022) for multilingual S2ST.

## 5 EXPERIMENTS

**Model Configurations.** For semantic representations, we apply XLSR-53 pre-trained on 56k hours of speech in 53 languages (Conneau et al., 2020) and use k-means to discretize 12th-layer embeddings

into semantic tokens with a codebook of size 1000 and a total downsampling rate of 320. For acoustic representation, we train the SoundStream model with 12 quantization levels, each with a codebook of size 1024 and the same downsampling rate of 320. We take three quantization levels as the acoustic tokens, representing each frame as a flat sequence of tokens from the first, second, and third quantization layers. We trained three sets of MVoice, with 160M (base), 520M (medium), and 1.2B (large) parameters. As for the unit-based vocoder, we use the modified V1 version of BigVGAN. A comprehensive table of hyperparameters is available in Appendix B. Except explicitly stated, we use our 520M (medium) model for downstream evaluation.

**Training.** During training, we train MVoice for 100K steps using 8 NVIDIA V100 GPUs with a batch size of 6000 tokens for each GPU on the publicly-available *fairseq* framework (Ott et al., 2019). Adam optimizer is used with $\beta_1 = 0.9, \beta_2 = 0.98, \epsilon = 10^{-9}$. $S_3$ model is optimized with a segment size of 8192 and a learning rate of $1 \times 10^{-4}$ until 500K steps using 4 NVIDIA V100 GPUs. For sampling, we employ top-p (Holtzman et al., 2019) sampling with p = 0.25.

## 5.1 MONOLINGUAL AND CROSS-LINGUAL ZERO-SHOT TEXT-TO-SPEECH

Table 4: Quality and style similarity of generated samples in monolingual zero-shot text-to-speech.

| Model | MOS ($\uparrow$) | SMOS ($\uparrow$) | WER ($\downarrow$) | SIM ($\uparrow$) |
|---|---|---|---|---|
| GT | 4.23±0.09 | / | 4.1 | / |
| GenerSpeech | 3.99±0.08 | 3.77±0.08 | 8.6 | 0.83 |
| YourTTS | 3.89±0.08 | 3.72±0.06 | 12.1 | 0.78 |
| MVoice | **4.04±0.07** | **3.81±0.08** | 6.7 | **0.85** |
| **Small-Scale Subjective Test** | | | | |
| VALL-E | 3.92±0.12 | 3.81±0.07 | 4.5 | 0.79 |
| MVoice | 4.01±0.06 | 3.87±0.04 | 3.0 | 0.83 |

1) For the intelligibility of the generated speech, MVoice has achieved a WER of 6.7, comparable with other systems, indicating that MVoice could generate accessible speech of good quality as previous non-autoregressive TTS families. 2) For audio quality, MVoice has achieved the highest MOS with scores of $4.04$ compared with the baseline models, demonstrating the effectiveness of the vocoder in generating high-fidelity waveforms. 3) Regarding style similarity, MVoice scores the SIM of $0.85$, showing that MVoice surpasses the state-of-the-art models in transferring the style of custom voices. Informally, MVoice is optimized in a large amount of self-supervised data, which contains many speakers with various accents, diverse demographics, and heterogeneous recording conditions, to improve robustness and generalization in zero-shot scenarios.

Using the examples provided on its demo page, we also compare MVoice with VALL-E in a small-scale subjective test. We synthesize utterances using the same transcripts and prompts and conduct the objective and subjective test with the same protocol described above. Table 4 shows that, in these examples, MVoice obtains 1.5 lower WER and 0.04 higher style similarity than baseline models in zero-shot synthesis.

Table 5 presents cross-lingual zero-shot TTS results, where the audio context and the target text are in different languages. For each target text, we sample one 3-second-long audio context from each language, which creates language transfer directions in total. Compared with YourTTS, MVoice yields better results in most languages, obtaining lower WERs and higher similarity SIM averaged across audio contexts. Regarding low-resource language, MVoice presents potential improvement for the limited usage of training data at this time.

## 5.2 SINGING VOICE SYNTHESIS

Table 6 demonstrates that MVoice (SVS) outperforms the baseline system by a large margin in terms of pitch similarity, showing distinct 70%/30% superiority over FFT-Singer/DiffSinger in terms of FFE objective evaluation. MVoice can resemble the note prompt and demonstrates its precise pitch reconstruction. Regarding singer similarity, MVoice scores the highest SIM of $0.78$, surpassing the state-of-the-art models in transferring the style of custom singing voices in zero-shot scenarios even though the voice used for prompting is never seen at training.

Table 5: Quality and style similarity of generated samples in multilingual zero-shot text-to-speech. YT refers to YourTTS. We report SIM for simplification for the Fr, Zh, and Ja languages.

| | Prompt | De WER | De SIM | En WER | En SIM | Es WER | Es SIM | Fr SIM | Zh SIM | Ja SIM |
|---|---|---|---|---|---|---|---|---|---|---|
| YT | De | 6.0 | 0.81 | 3.1 | 0.71 | 4.1 | 0.71 | 0.72 | 0.74 | / |
| | En | 8.0 | 0.79 | 6.3 | 0.71 | 3.0 | 0.78 | 0.71 | 0.72 | / |
| | Es | 10.3 | 0.71 | 2.6 | 0.73 | 12.3 | 0.80 | 0.70 | 0.70 | / |
| | Fr | 12.7 | 0.76 | 5.1 | 0.71 | 18.6 | 0.67 | 0.79 | 0.67 | / |
| | Zh | 21.3 | 0.72 | 11.0 | 0.65 | 2.0 | 0.75 | 0.76 | 0.75 | / |
| | Ja | 6.1 | 0.80 | 10.1 | 0.73 | 2.1 | 0.69 | 0.82 | 0.79 | / |
| | **AVG** | **10.7** | **0.76** | 6.3 | 0.70 | **7.0** | 0.73 | 0.75 | 0.72 | / |
| Ours | De | 10.1 | 0.78 | 4.2 | 0.75 | 14.1 | 0.75 | 0.78 | 0.72 | 0.70 |
| | En | 15.1 | 0.78 | 9.1 | 0.77 | 9.1 | 0.80 | 0.74 | 0.79 | 0.67 |
| | Es | 13.0 | 0.76 | 7.1 | 0.75 | 13.0 | 0.78 | 0.78 | 0.68 | 0.70 |
| | Fr | 22.0 | 0.70 | 3.6 | 0.71 | 11.0 | 0.73 | 0.78 | 0.77 | 0.69 |
| | Zh | 10.3 | 0.68 | 5.3 | 0.71 | 18.1 | 0.69 | 0.76 | 0.70 | 0.68 |
| | Ja | 9.1 | 0.79 | 8.0 | 0.77 | 8.1 | 0.85 | 0.80 | 0.76 | 0.93 |
| | **AVG** | 13.2 | 0.75 | **6.2** | **0.74** | 12.2 | **0.77** | **0.77** | **0.74** | **0.72** |

## 5.3 VOICE CONVERSION AND SINGING VOICE CONVERSION

Table 6: SVS. Note that FFT-Singer and Diffsinger conduct in-domain generation with seen speaker while MVoice presents zero-shot SVS.

| Model | MOS (↑) | SMOS (↑) | SIM (↑) | FFE (↓) |
|---|---|---|---|---|
| GT | 4.08±0.08 | / | / | / |
| FFT-Singer | 3.86±0.05 | 3.91±0.08 | 0.66 | 0.12 |
| Diffsinger | 3.96±0.07 | 3.94±0.07 | 0.67 | 0.11 |
| MVoice (Zero-shot) | **3.99±0.06** | **3.96±0.05** | **0.78** | **0.08** |

Table 7: Zero-shot VC and SVC.

| Model | MOS (↑) | SMOS (↑) | SIM (↑) |
|---|---|---|---|
| **Voice Conversion** | | | |
| Prompt | 4.26±0.06 | / | / |
| NANSY | 3.89±0.08 | 3.73±0.10 | 0.68 |
| PPG-VC | 3.97±0.06 | **3.82±0.05** | 0.78 |
| MVoice (Zero-shot) | **4.02±0.08** | 3.78±0.06 | **0.80** |
| **Singing Voice Conversion** | | | |
| Prompt | 4.21±0.05 | / | / |
| MVoice | 3.96±0.06 | 3.72±0.05 | **0.76** |

Table 7 shows that MVoice scores the comparable overall SIM of 0.93 with baseline. It excels at converting speaker identity even in a zero-shot scenario, attributing to the scalable training data covering diverse speakers with various accents. For audio quality, it presents high perceptual quality with outperformed MOS evaluation. To conclude, MVoice converts the timbre with better naturalness and comparable speaker similarity to baseline models, even though the model is trained without any text transcript paired with audio recordings. For singing voice conversion (SVC), MVoice also excels at converting singer identity and presents good perceptual quality and naturalness.

## 5.4 SPEECH-TO-SPEECH TRANSLATION

Table 8 shows a decrease in translation accuracy compared with state-of-the-art models in Speech-Matrix datasets. Informally, a gap from baseline models regarding translation accuracy could be witnessed: Different from straight-forward semantic mapping in voice synthesis task, it is thus perhaps not surprising that a decrease in performance on the speech translation tasks could be witnessed, since model capacity must be devoted to semantic translation and conversion simultaneously.

For speech quality, since we apply the publicly available pre-trained unit vocoder, which mainly controls the naturalness of output speech and leaves it unchanged, we expect S2ST to exhibit high-quality speech generation as baselines.

## 5.5 ANALYSIS AND ABLATION STUDIES

To verify the effectiveness of several designs in MVoice, we conduct ablation studies and discuss the key findings as follows. In this section, we first analyze the model scalability, then qualitatively

Table 8: BLEU score in S2ST. A decrease in the speech translation tasks could be witnessed since model capacity must be devoted to semantic translation (S2ST) and conversion (TTS/SVS) simultaneously.

| Model | En-Es | En-De | En-Fr |
|---|---|---|---|
| Hours | 1366 | / | 1518 |
| SpeechMatrix | 21.9 | 10.1 | 19.2 |
| MVoice | 16.3 | 7.3 | 12.3 |

Table 9: We compare with 1) different scales and 2) multilingual (M) or monolingual training data.

| Model | TTS-WER | TTS-SIM | VC-SIM |
|---|---|---|---|
| Base | 9.8 | 0.84 | 0.75 |
| Medium | 8.3 | 0.86 | **0.76** |
| Large | **6.1** | **0.87** | **0.76** |
| De | 18.0 | 0.65 | 0.52 |
| De (M) | **10.1** | **0.78** | **0.72** |
| Fr | 44.0 | 0.53 | 0.62 |
| Fr (N) | **30.2** | **0.72** | **0.79** |

investigate the benefits of multilingual and multitask training, and finally explore the ability to maintain voice emotion and noise continuations.

**Model Scalability.** Table 9 reports LJSpeech results for different model sizes, namely 160M (base), 520M (medium), and 1.2B (large) parameter models. As expected, scaling the model size results in better scores. However, this comes at the expense of longer training and inference time. Increasing the model size further from 520M to 1.2B leads to additional gains of a further 40% reduction in WER for TTS tasks with a similar style similarity.

**Multilingual and Multitask.** The limited amount of data is a significant challenge for low-resource languages. For one model, we train with monolingual speech tasks; for the other, we include the combined task of 6 languages. MVoice leverages a joint vocabulary and trains a single decoder-only model on a mixture of tasks that involve arbitrarily multilingual voice, and Table 9 shows the improved performance with the combination of simpler tasks. Combined language and task training reduce the problem to a pipeline approach of separate systems, which helps the model to better connect its semantic understanding to its acoustic audios especially in low-resource languages, leading to the gains of 7.9 reduction in WER (De) for TTS tasks and a 0.2 point improvement in SIM (De).

**Zero-shot transfer beyond speaker identity.** This section presents how our approach could be extended beyond, including cross-lingual timbre transferring, generating coherent emotion, and noise continuations. We have attached the information on testing data in Appendix A. As shown in the demo page, we find that 1) MVoice can preserve the **emotion** in the prompt at a zero-shot setting, even if the model is not fine-tuned on an emotional TTS dataset; 2) MVoice effectively reproduces the characteristics from a **cross-lingual** style prompt, which has not been seen during training; and 3) In a noisy environment, the model also presents the acoustic consistency and maintain the **noise conditions** in the prompt.

## 6 CONCLUSION

In this work, we proposed MVoice, a multimodal large language model for synthesizing and manipulating multilingual voice signals at scale. MVoice enjoyed the "coarse-to-fine" design to effectively model the human voice by considering semantic meanings and acoustic conditions. Experimental results demonstrated that MVoice achieves state-of-the-art results in monolingual/cross-lingual zero-shot TTS and comparable results in speech-to-speech translation. MVoice offered notable advantages as a general-purpose unified model in voice synthesis: 1) model and data scalability: without the requirement of annotations acoustic data or scattered model-specific methodologies, training could be scaled up in terms of data usage and model capability; 2) controllability and conditioning flexibility: various conditioning mechanisms were investigated by re-synthesizing the semantic or acoustic representations with prompt guidance. For future work, we will verify the effectiveness in more general scenarios of visual modality. We envisage that our work will serve as a basis for future voice synthesis studies.

**Limitation.** Although MVoice as a spoken large language model is successfully applied to multilingual zero-shot voice signals at scale, it still suffers from some limitations: 1) MVoice introduces a strong dependency on the quality of the audio tokenizer. 2) The model only shows in-context learning ability on voice synthesis, rather than all voice recognition and understanding tasks, and 3) a longer sequence length typically requires more computational resources, and degradation could be witnessed with decreased training data.

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
