# Appendices

# MVoice: Unified Voice Synthesis With Discrete Representation

## A    DATA

In this section, we describe details of the data usage in training and evaluating MVoice.

- SpeechMatrix (Jia et al., 2022), a large-scale multilingual corpus of speech-to-speech translations mined from real speech of European Parliament recordings. We use the benchmark dataset for speech-to-speech translation experiments.
- Common Voice (Ardila et al., 2019) consists of text paired with recordings where people were asked to read the text aloud.
- Librilight (Kahn et al., 2020) contains 60K hours of unlabelled speech from audiobooks in English, and LibriSpeech (Panayotov et al., 2015), LibriTTS (Zen et al., 2019), Gigaspeech (Chen et al., 2021), AISHELL (Fu et al., 2021), VCTK (Veaux et al., 2017) datasets include transcriptions.
- CSD (Choi et al., 2021b) contains multilingual singing voice. We also use the female-singer OpenCPOP (Wang et al., 2022), multi-singer dataset OpenSinger (Huang et al., 2021), and M4Singer (Zhang et al., 2022) as the singing voice data.

## B    MODEL CONFIGURATIONS

We list the model hyper-parameters of MVoice in Table 10.

| Hyperparameter | | MVoice |
|---|---|---|
| MVoice Global Base | Transformer Layer | 16 |
| | Transformer Embed Dim | 768 |
| | Transformer Attention Headers | 12 |
| | Number of Parameters | 114 M |
| MVoice Global Medium | Transformer Layer | 20 |
| | Transformer Embed Dim | 1152 |
| | Transformer Attention Headers | 16 |
| | Number of Parameters | 320 M |
| MVoice Global Large | Transformer Layer | 24 |
| | Transformer Embed Dim | 1536 |
| | Transformer Attention Headers | 32 |
| | Number of Parameters | 930 M |
| MVoice Local | Transformer Layer | 6 |
| | Transformer Embed Dim | Same as global |
| | Transformer Attention Headers | 8 |
| | Number of Parameters | 46/101/303 M |
| BigVGAN Vocoder | Upsample Rates | [5, 4, 2, 2, 2, 2] |
| | Hop Size | 320 |
| | Upsample Kernel Sizes | [9, 8, 4, 4, 4, 4] |
| | Number of Parameters | 121.6M |

Table 10: Hyperparameters of MVoice.

## C    APPLICATIONS

### C.1    MIDI-TO-F0 CONVERTER

Singing voice synthesis (SVS) is a task that generates singing voices from the given music score and lyrics like human singers. Following (Liu et al., 2022; Zhang et al., 2022), the SVS system typically

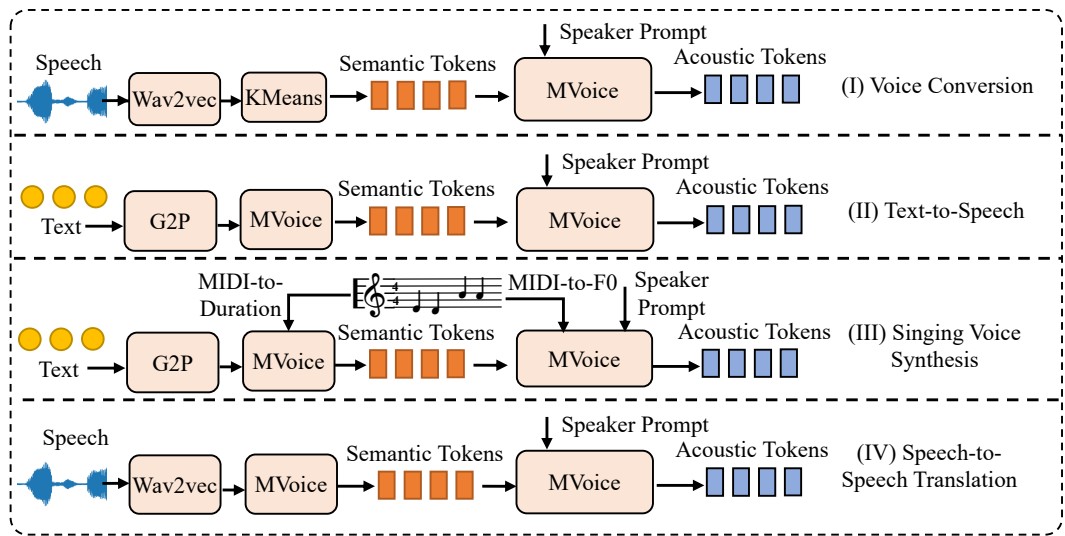

Figure 2: We introduce 5 exemplar applications, including voice conversion (VC), text-to-speech (TTS), singing voice synthesis (SVS), singing voice conversion (SVC), and speech-to-speech translation (S2ST), that can be tackled by sharing a voice synthesis framework with semantic and acoustic tokens.

includes the MIDI-to-F0 converter to predict F0 explicitly. Though the SVS system can be further improved with the direct MIDI condition and implicit F0 prediction, this is beyond our focus.

## C.2  UNIT-BASED VOCODER

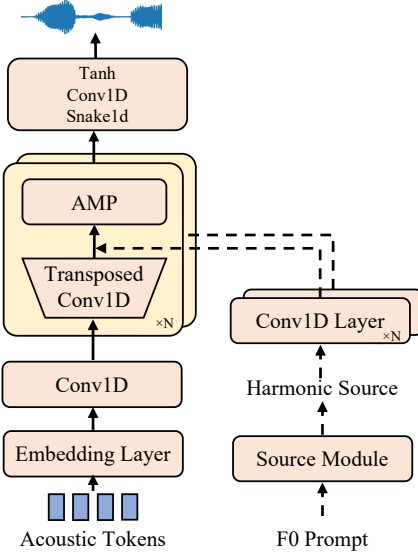

Figure 3: Overview of the unit-based vocoder. The F0 auxiliary input denoted with dotted lines is included only in singing voice synthesis.

The generator of the unit-based vocoder is built from a set of look-up tables (LUT) that embed the discrete representation, and a series of blocks composed of transposed convolution and a residual block with dilated layers. We train the enhanced vocoder with the weighted sum of the least-square adversarial loss, the feature matching loss, and the spectral regression loss on mel-spectrogram, where the training objective formulation and hyperparameters follow Kong et al. (2020); Lee et al. (2022).

For speech generation, we train the vocoder with only the discrete unit sequences as input. For singing voice generation, we further include F0-driven source excitation to stabilize long-continuous waveforms generation following (Liu et al., 2022; Huang et al., 2022a).

As illustrated in Table 11, replacing the unit-based vocoder with a SoundStream decoder for voice synthesis has witnessed a distinct degradation of perceptual quality, proving that the codebook mismatch for the SoundStream decoder between training (12 quantization levels) and inference (3 levels) hurts reconstruction performance. In contrast, a neural vocoder could refine the coarse-grained acoustic tokens and generate waveforms with increasing details.

Table 11: Ablation studies.

| Model | STOI ($\uparrow$) | MCD ($\downarrow$) |
|---|---|---|
| $S_3$: SoundStream | 0.92 | 1.90 |
| $S_3$: Unit Vocoder | **0.93** | **1.56** |

## D    EVALUATION

### D.1    SUBJECTIVE EVALUATION

For audio quality evaluation, we conduct the MOS (mean opinion score) tests and explicitly instruct the raters to "*(focus on examining the audio quality and naturalness, and ignore the differences of style (timbre, emotion, and prosody).)*". The testers present and rate the samples, and each tester is asked to evaluate the subjective naturalness on a 1-5 Likert scale.

For style similarity evaluation, we explicitly instruct the raters to "*(focus on the similarity of the style (timbre, emotion, and prosody) to the reference, and ignore the differences of content, grammar, or audio quality.)*". In the SMOS (similarity mean opinion score) tests, we paired each synthesized utterance with a ground truth utterance to evaluate how well the synthesized speech matches that of the target speaker. Each pair is rated by one rater.

Our subjective evaluation tests are crowd-sourced and conducted by 20 native speakers via Amazon Mechanical Turk. The screenshots of instructions for testers have been shown in Figure 4. We paid $8 to participants hourly and totally spent about $600 on participant compensation. A small subset of speech samples used in the test is available at `https://MVoice.github.io/`.

### D.2    OBJECTIVE EVALUATION

Cosine similarity is an objective metric that measures speaker similarity among multi-speaker audio. We compute the average cosine similarity between embeddings extracted from the synthesized and ground truth embeddings to measure the speaker similarity performance objectively.

Word Error Rate (WER) evaluates the faithfulness to the input transcript by transcribing the synthesized utterances using a wav2vec ASR system.

F0 Frame Error (FFE) combines voicing decision error and F0 error metrics to capture F0 information.

Mel-cepstral distortion (MCD) measures the spectral distance between the synthesized and reference mel-spectrum features.

Short-time objective intelligibility (STOI) assesses the denoising quality for speech enhancement.

## E    POTENTIAL NEGATIVE SOCIETAL IMPACTS

MVoice lowers the requirements for high-quality and expressive text-to-speech synthesis, which may cause unemployment for people with related occupations such as broadcaster and radio host. In addition, there is the potential for harm from non-consensual voice cloning or the generation of fake media and the voices of the speakers in the recordings might be over-used than they expect.

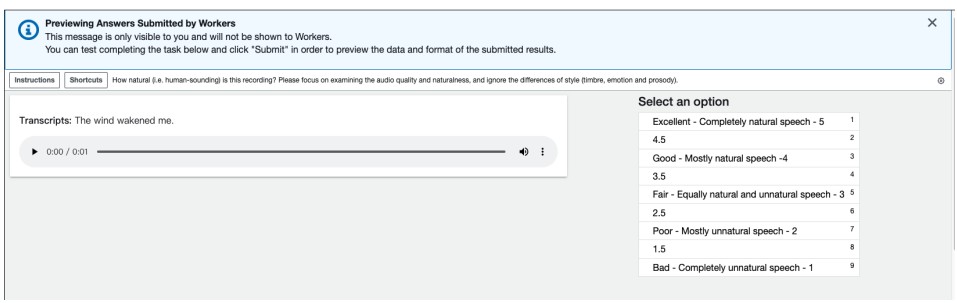

(a) Screenshot of MOS testing.

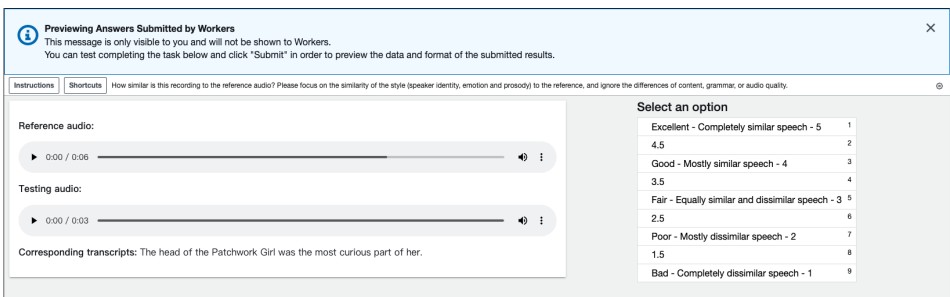

(b) Screenshot of SMOS testing.

Figure 4: Screenshots of subjective evaluations.