# OpenReview forum: "MVoice: Multilingual Unified Voice Generation With Discrete Representation at Scale"
_ICLR.cc/2024/Conference — ICLR 2024 Conference Withdrawn Submission_

### Official Review · Reviewer_KBUu · 2023-11-01

**Soundness:** 2 fair
**Presentation:** 2 fair
**Contribution:** 2 fair
**Rating:** 3
**Confidence:** 3

**Summary:**

This paper introduces a generative model for speech called MVoice. It is a cascade of a semantic (text or speech to semantic tokens) and an acoustic module (semantics to waveform).

The model uses AR, transformer based model. The audio model is trained using in-context learning in a self-supervised way. The synthesized output is controllable through speaker representation, F0, etc. prompts. The model can be multilingual and it can support Voice Conversion, TTS, singing voice synthesis and conversion, and speech to speech translation.

The input units to the model are multilingual phonemes. After the audio feature generation, a BigVGAN vocoder is used to compute the audio signal. Training the semantic part requires parallel speech or text - semantic content data, acoustic stage is self-supervised.

In the experiments, the model is trained on 6 languages. Generated outputs are evaluated based on WER, speaker similarity to the prompt and MOS from a set of crow-sourcers. In monolingual TTS, the proposed method is marginally better than GenerSpeech. In the multilingual setting, it cannot outperform YourTTS in terms of WER even though it achieves better speaker similarity scores. In singing voice synthesis, MVoice is better than Diffsinger. On speech translation, SpeechMatrix outperforms MVoice.

Some limitations of the model has been mentioned at the end of the paper.

**Strengths:**

- Originality:
Even though discrete unit-based generative models been investigated before (e.g., Polyak et al. 2021, https://arxiv.org/pdf/2104.00355.pdf), this paper adds AR modeling and controllability to obtain a more unified model in terms of the variety of that can be performed with the same model. Some advantages of the model are
1. Experiments usually achieve good performance on several speech tasks as compared to the previously published studies.
2. Being controllable is an advantage of the model to adapt it to different use cases.
3. The modularity (semantic stage and then an acoustic stage) allows the model to perform different tasks.

- Quality:
The paper proposed a two-stage speech model (semantic + acoustic models) which makes it somewhat interpretable. Experiments have been conducted on various tasks and the performance of those are on par with that of previous studies if not better.

- Clarity:
At a high-level, the paper is relatively easy to understand, but in terms of details, some parts have been left to the Appendix and it might make the paper difficult to read at times.

- Significance:
Similar to the comments in Originality above. This is a study towards investigating unified generative speech models.

**Weaknesses:**

1. Experiments do not always show gain, and when they outperform, the improvements are limited.

2. Even though the paper title mentions multilingual unified speech representations, the results on multilingual condition are worse than the monolingual ones.

3. Overall, the main gains are coming from improved speaker similarity scores and not in terms of other evaluation criteria.

4. The paper already mentions a few shortcomings and these are valid concerns. Regarding the dependency of the performance on the tokenizer quality makes the reason for using discrete tokens questionable. If there were continuous features, would it be better? If so, then wouldn't the first stage lose its property, i.e. being interpretable as semantic stage?

**Questions:**

1. If there were continuous features, would the model be better? If so, then wouldn't the first stage lose its property, i.e. being interpretable as semantic stage?

2. Which ASR models have been used to obtain the WERs on the synthetic datasets?

---

### Official Review · Reviewer_B8JJ · 2023-11-04

**Soundness:** 3 good
**Presentation:** 2 fair
**Contribution:** 1 poor
**Rating:** 3
**Confidence:** 5

**Summary:**

This paper studies the tasks of speech and voice synthesis under several settings (text-to-speech, voice conversion, singing voice synthesis and conversion,  and speech-to-speech translation). The authors propose MVoice, a cascade of Transformer decoder models, operating in a "coarse-to-fine" manner. MVoice first generates the "coarse" units, which mainly capture the "content" to be generated, and then it predicts the "fine" units, which capture the fine-grained acoustic details. The authors additionally introduce another level of controllability during the "fine" unit generation in the form of an acoustic prompt or F0.
The authors compared the proposed method to several baseline methods and showed that their approach reached superior performance. Lastly, the authors present an analysis and ablation study analyzing model scalability, multi-lingual and multi-task learning, and zero-shot capabilities.

**Strengths:**

1. MVoice was trained using a multi-task and multi-lingual objective
2. MVoice introduces an additional conditioning to the "fine" units prediction
3. The authors present results and experiments on various tasks, including TTS, VC, SVS, SVC, and S2ST
4. The authors present results for models at different scales, and provide analysis and a small ablation study

**Weaknesses:**

1. The proposed method is not much different than prior work such as [1]. MVoice, using the exact same modeling while applying multi-lingual and multi-task objectives. Other differences are very minor and incremental.
2. The paper is not written well. There are some notation issues, missing details, and references. It would be very hard for someone not 100% familiar with the current literature in the field to understand the paper and the details. For example, under the "Acoustic tokens" paragraph, it is not clear that there are several codebooks. The notation of the sub-script is also confusing "The residual vector-quantizer Q produces discrete representations $a_q$ ..." and later on, the authors refer to this sub-script as time and not codebook.
3. The sample page is restricted, hence we can not listen to the samples.

[1] Borsos, Zalán, et al. "Audiolm: a language modeling approach to audio generation." IEEE/ACM Transactions on Audio, Speech, and Language Processing (2023).

**Questions:**

1. In section 3.3.2 the authors state that $S_2$ "is conditioned on the condition signals c_a (semantic tokens, acoustic prompts, or explicit F0)". Can $S_2$ not be conditioned on the semantic tokens? In that case, $S_1$ is not needed.
2. Did the authors try to manipulate the F0?
3. The authors claim that $S_2$ can be trained in an unsupervised manner and, hence can serve as a foundational model. Did the authors try to quantify that empirically? How much data is needed? what is the effect of adding more data for this stage of training?
4. The authors replaced their decoder with a unit based-BigVGAN, what is the reason for that? how do the results look like / sound like with the original decoder?
5. When trying to reach the samples shared by the authors we get a 404 error, hence can not listen to the samples.

---

### Official Review · Reviewer_c2w8 · 2023-11-05

**Soundness:** 3 good
**Presentation:** 2 fair
**Contribution:** 3 good
**Rating:** 5
**Confidence:** 4

**Summary:**

This work proposes MVoice, a multilingual model that unifies the training of a generative model for multiple tasks including text-to-speech, voice conversion, singing voice synthesis and conversion, and speech-to-speech translation. The paper breaks down the training into two steps following a coarse-to-fine modeling approach. In the first stage, semantic mapping is learned from text/speech using XLSR-53 as the tokenizer for targets. The subsequent stage learns to produce fine-grained acoustic tokens with semantic tokens, acoustic prompts, and optional F0 as the conditioning input. Finally, a unit-based neural vocoder maps acoustic tokens to waveforms. Decoupling into two stages allows for scaling the second stage to large amounts of unlabelled data and for zero-shot applications via the correct specification of task and language-specific tags. Experiments are conducted to showcase the viability of this approach for the aforementioned tasks and out-of-distribution settings.

**Strengths:**

- The two-stage approach of mapping to semantic tokens followed by a fine-tuning stage has been explored in literature before, such as with AudioLM and SpearTTS, but scaling it to incorporate multiple tasks through context specification tags is novel and shows significant promise.
- The paper conducts exhaustive experiments on a variety of tasks to showcase the the versatility and efficacy of MVoice.

**Weaknesses:**

- The link to the audio examples is broken. I attempted to access it using both Mozilla and Chrome browsers but both failed.
- Missing details and unclear writing:
  - While the proposed method is novel, the paper often lacks clarity, may have overlooked some important details, or uses ambiguous language. I have used the following to refer to a specific line: (P1, P2, L3) indicates Page 1, Paragraph 2, Line 3. I list several such instances below.
    - Abstract, (P2, P3, L1): The phrase "without the requirement of scattered model-specific methodologies" is unclear. The meaning does not become clear until the Related Works section (P3, P1-2). It would be beneficial to explain this in a sentence earlier to clarify the writing. Given the prior works that scale to unlabelled data, it would also be advantageous to make the contribution more specific regarding multi-task training of stage 1 and scaling to more applications.
    - Abstract, (P2, P3, L4), (P3, P3, L9): "we investigate different conditioning mechanisms ..." should be clarified early in the paper.
    - Abstract, (P2, P4, L5): Experimental results details are missing: what metrics are used, and by how much do they improve?
    - (P2, P1, L5): The discussion about "unified generation: most of current ... scattered in research fields" is too vague regarding methodologies or applications.
    - (P2, P5, L1):  Could you please provide clarification on the term 'multimodal' in the context of this work?
- Languages and tasks module:
  - The steps are unclear. Is it the following? For each stage, S1 and S2, appropriate tags are provided as a prefix. For example:
    - For S1 and the task S2ST, assuming parallel data for source A and target T, would the input be:  *[S2ST] wav2vec tokens (A)*, and the target for AR be: *wav2vec tokens (T)*?
    - Similarly, for S2, are the tasks [S2A] or [S2AwithF0] for semantic to acoustic without and with F0 respectively? How is F0 integrated into the second stage, S2?

- Missing details on the metrics:
  - Speaker similarity: Which model is used for measuring similarity? Notably, [2] reports much lower speaker similarity scores using the WavLM-TDCNN speaker embedding model on the YourTTS model. The WER reported by [2] using the Whisper large-v2 model also appears much lower.
    - Is the speaker similarity calculated between the original prompt or the vocoded prompt?
  - Following Speech-X, Vall-E, Voicebox, and SpearTTS, using [3] for speaker similarity evaluations is recommended.
  - Thank you for referencing [1] regarding multilingual ASR models, it would also be beneficial to include all models in the Appendix for a self-contained paper.

- Potentially incorrect claim:
  -  Considering the results in Table 5, I question the claim of the paper at (P7, P5, L3): "Compared with YourTTS, MVoice yields better results in most languages, obtaining lower WERs and higher similarity scores averaged across audio contexts." In 13 out of 18 settings, the WER is higher than that of the YourTTS model.

[1] Speechmatrix: A large-scale mined corpus of multilingual speech-to-speech translations.

[2] Voicebox: Text-Guided Multilingual Universal Speech Generation at Scale

[3] Wavlm: Large-scale self-supervised pre-training for full stack speech processing.

**Questions:**

- Speech-X also supports other tasks such as noise suppression, speech removal, target speaker extraction, clean speech editing, and noisy speech editing. Kinldly clarify if MVoice supports this? If not, it would be appropriate to acknowledge this difference.
- Do stages S1 and S2 use the same model with appropriate prefixing, or are they two different models?
- The figures in the appendix explaining the applications are appreciated. Pointing them out in the main paper would be beneficial.
- (P2, P2, L5): "The single decoder-only model ... interleaved voice": A figure would greatly help clarify this point.

---

### Official Review · Reviewer_at49 · 2023-11-07

**Soundness:** 2 fair
**Presentation:** 2 fair
**Contribution:** 2 fair
**Rating:** 3
**Confidence:** 4

**Summary:**

This paper describes MVoice, a multimodal decoder-only speech language model which operates in the space of a set of discrete self-supervised speech representations. The model is trained to carry out multiple different tasks across 6 languages, namely speech synthesis and voice conversion for both regular speech and singing, and direct speech-to-speech translation. Depending on the specific task, the model takes some combination of phoneme sequences, discrete speech token sequences and raw speech audio prompts as input, plus special tokens indicating the desired task and target language. The model outputs discrete speech tokens at one of two levels: "semantic" tokens which represent the underlying phonetic content of the output speech utterance, and "acoustic" tokens which represent additional acoustic detail such as speaker timbre, prosody and recording conditions. To generate an output speech utterance, the model is run in two stages to match this hierarchical representation of speech, first generating semantic tokens and then refining the semantic token sequence to produce an acoustic token sequence, which is then passed to a neural vocoder trained to convert acoustic tokens to waveforms.

**Strengths:**

The main strength of the paper is its treatment of multiple speech generation tasks within a single model, allowing a unified prompt-based interface to select the desired task and target language for output speech. This paper also appears to be the first to include singing speech synthesis and voice conversion in such a design.

The authors show convincing and competitive results against sensible baselines for zero-shot (i.e. for speakers unseen during training) speech synthesis and voice conversion in English, as measured by subjective ratings of speech quality and naturalness, speaker similarity and intelligibility. Singing voice synthesis results for an unseen Mandarin speaker also appear strong.

**Weaknesses:**

The paper fails to distinguish itself sufficiently from recent prior work. For example, the authors reference SPEAR-TTS which also operates in two stages, first predicting "semantic" tokens from text, which are then used to generate an "acoustic" token sequence which can be converted to audio. As in MVoice, acoustic token prediction in SPEAR-TTS is conditioned on a short audio prompt which provides speaker timbre and other such acoustic conditions for the output audio. The other major claim to novelty in the paper is prompt-based task specification, where a single MVoice model can perform many tasks, selected by special tags provided as part of the input. This approach has recently been presented in AudioPaLM [1], albeit for a slightly different selection of tasks. This seems to leave the inclusion of (singing) voice conversion in the training tasks as the main distinguishing feature of MVoice over these prior works. However, it doesn't seem to me that there is anything particular in the model setup which enables voice conversion in MVoice which AudioPaLM could not also do -- only those prior authors decided not to include that task. Moreover, results presented for (singing) voice conversion and singing voice synthesis show much smaller differences from prior work compared to the monolingual TTS results, so that this main innovation seems of limited impact.

Voice conversion also seems to me to be the source of the claim that MVoice is better able to scale to larger data sets because it avoids the "requirement of annotations [sic] audio data", as compared to VALL-E and SPEAR-TTS. This is because voice conversion can be performed given only semantic and acoustic tokens from source and target audio, without text transcripts (where semantic tokens stand in as a strong proxy). The other tasks in MVoice, TTS and speech-to-speech translation, otherwise require much stronger annotation, respectively text transcripts and parallel audio in two languages. I think perhaps more focus should have been put on voice conversion and singing voice synthesis to properly distinguish this paper from other recent works.

Many details are not clear from the current presentation in the paper, especially with regard to evaluation, which makes it difficult to justify the authors' claims. Table 3 lists LibriTTS and VCTK as the testing corpora for TTS and VC tasks; these corpora are English only, so it's unclear what data was used to evaluate zero-shot cross-lingual TTS with target languages other than English (Section 5.1, Table 5). It's also unclear whether the monolingual TTS results in Section 5.1, Table 4 are English-only or represent performance across multiple languages. These details should be clarified, along with the balance of languages in the combined training data (at least in terms of total duration), to give proper context to these results.

The discussion of certain results in the main text also does not match the numbers presented in results tables, particularly for cross-lingual TTS. In the final paragraph of Section 5.1, the authors state that "compared with YourTTS, MVoice yields better results in most languages, obtaining lower WERs and higher similarity SIM". However, Table 5 shows WER results only for 3/6 languages; of these, English shows a very small WER improvement while German and Spanish show that MVoice is relatively much worse that YourTTS. WER results for French, Chinese and Japanese are omitted from this table "for simplification", although Table 9 seems to show a TTS-WER rate for French (from a multilingual model) of 30.2, even higher than those shown in Table 5 (I believe this figure corresponds with those in Table 5 because the German result of 10.1 matches). The reported SIM values (often referred to in the text as "style similarity", although I think really it is speaker similarity) show only small improvements on YourTTS, but I would consider intelligibility as measured by WER to be the more significant metric in this context, and would prefer to see those numbers. I also think that some subjective evaluation is needed here as well. Given all of this, I am quite sceptical of the claims that MVoice provides better overall performance in this cross-lingual TTS context than YourTTS.

[1] Rubenstein et al. (2023) AudioPaLM: A Large Language Model That Can Speak and Listen, arXiv:2306.12925

**Questions:**

On the choice of semantic tokens:
- How did you select which layer of the XLSR-53 to extract representations from?
- Did you use the Base or Large XLSR-53 model?
- How did you choose the number of k-means clusters to use?
- Does multilingual training need more clusters than monolingual training? Was your choice of 1000 clusters aimed at one of these settings in particular?
- Is there any reason not to use the internal codebooks learned during XLSR-53 pre-training?
- If these choices were influence by prior literature, this should be mentioned in the text.

Somewhat related, it seems from Section 3.6 "Textless S2ST" that speech prompts for S2ST (and VC?) are extracted using another Wav2Vec 2.0 model, which I then assume is trained on English LibriSpeech only? Again, which pre-trained model did you use, and which layer did you extract continuous representations from? Why not also use the XLSR-53 model here? More details should be given on this, perhaps earlier in the paper, since for a long time I didn't understand how the speech/speaker prompts shown in Figure 1 were being generated.

In Section 3.6 "Zero-shot TTS / VC", you say that "two non-overlapping windows of speech from each example" are used as prompt and target output. Is this only for VC training, of the S_2 semantic-to-acoustic token stage, where you already have frame-alignments from the fixed framerate of the tokens? I don't understand how this could be applied to TTS text-to-semantic tokens training without having frame-level phone alignments, which you don't mention explicitly anywhere (except for SVS S_1 "text-to-semantic with duration"). If you in fact need those alignments for training the TTS task, you should mention it somewhere.

In Section 5 "Model Configurations" you say "we train the SoundStream model with 12 quantization levels" and then "We take three quantization levels as the acoustic tokens". Do you mean that you trained your own SoundStream model from scratch? Why not train it with only 3 quantization levels in the first place? Did you try this, and find it not to work as well as training a separate BigVGAN vocoder on the first three levels of acoustic tokens? How did you decide that three levels was the best trade-off between final acoustic quality and sequence length? (I assume this is the limiting factor since you flatten the token sequences)

How many hours of data do you have for each language in the combined training data set? Do you expect this has any influence on any of your results?

Evaluation questions:
- What data do you use for evaluating TTS and VC in languages other than English? Table 3 does not show any non-English testing set for TTS/VC.
- Thank you for providing the question text and screenshots of your subjective evaluations! Still, some additional details are needed on the subjective evaluation procedure. You say you had "20 native speakers", is that for every language tested? How many utterances did each participant listen to? How many different utterances were evaluated in total?
- Same questions about the "small-scale subjective test" in Table 4 comparing MVoice and VALL-E -- it was really not clear to me how this differed from your main evaluation, or why you had to use this different test set.
- Where do the speaker embeddings used for SIM evaluation come from?
- I think you need to mention the significantly worse WER for French compared to German in Table 9 in the main text -- what do you think makes the difference here?

Speech-to-speech translation (Section 5.4):
- What does the / in the En-De/Hours cell of Table 8 mean? Is this an unseen language pair during training?
- I think you could be more explicit about the differences between MVoice and the SpeechMatrix baseline, especially with respect to the separate vocoder used in SpeechMatrix, and the fact that the SpeechMatrix speech-to-unit model essentially predicts "semantic" tokens rather than fine-grained acoustic tokens. Is that what you are trying to say in the second paragraph of Section 5.4?

Some relatively minor proof-reading points:
- The phrase "annotations acoustic data" appears several times, it seems to mean "annotated acoustic data".
- In the last paragraph of Section 5.1, the phrase "which creates language transfer directions in total" appears to be missing the actual number of directions.
- I suspect many bibliography entries need to be updated so that they do not reference pre-print versions. For example, the Common Voice corpus has an LREC 2020 paper, and is not only on arXiv, GenerSpeech was published at NeurIPS 2022.